# HER2-/HER3-Targeting Antibody—Drug Conjugates for Treating Lung and Colorectal Cancers Resistant to EGFR Inhibitors

**DOI:** 10.3390/cancers13051047

**Published:** 2021-03-02

**Authors:** Kimio Yonesaka

**Affiliations:** Department of Medical Oncology, Kindai University Faculty of Medicine, 377-2 Ohno-Higashi Osaka-Sayamashi, Osaka 589-8511, Japan; yonesaka@med.kindai.ac.jp; Tel.: +81-72-366-0221; Fax: +81-72-360-5000

**Keywords:** EGFR, HER2, HER3, ADC, trastuzumab deruxtecan (T-DXd), patritumab deruxtecan, NSCLC, CRC, HNSCC, EGFR-TKI

## Abstract

**Simple Summary:**

Epidermal growth factor receptor (EGFR) is one of the anticancer drug targets for certain malignancies including nonsmall cell lung cancer (NSCLC), colorectal cancer (CRC), and head and neck squamous cell carcinoma. However, the grave issue of drug resistance through diverse mechanisms persists. Since the discovery of aberrantly activated human epidermal growth factor receptor-2 (HER2) and HER3 mediating resistance to EGFR-inhibitors, intensive investigations on HER2- and HER3-targeting treatments have revealed their advantages and limitations. An innovative antibody-drug conjugate (ADC) technology, with a new linker-payload system, has provided a solution to overcome this resistance. HER2-targeting ADC trastuzumab deruxtecan or HER3-targeting ADC patritumab deruxtecan, using the same cleavable linker-payload system, demonstrated promising responsiveness in patients with HER2-positive CRC or EGFR-mutated NSCLC, respectively. The current manuscript presents an overview of the accumulated evidence on HER2- and HER3-targeting therapy and discussion on remaining issues for further improvement of treatments for cancers resistant to EGFR-inhibitors.

**Abstract:**

Epidermal growth factor receptor (EGFR) is one of the anticancer drug targets for certain malignancies, including nonsmall cell lung cancer (NSCLC), colorectal cancer (CRC), and head and neck squamous cell carcinoma. However, the grave issue of drug resistance through diverse mechanisms persists, including secondary EGFR-mutation and its downstream RAS/RAF mutation. Since the discovery of the role of human epidermal growth factor receptor 2 (HER2) and HER3 in drug resistance, HER2- or HER3-targeting treatment strategies using monoclonal antibodies have been intensively examined and have demonstrated impressive responsiveness and limitations. Finally, an innovative targeted therapy called antibody drug conjugates (ADC) has provided a solution to overcome this resistance. Specifically, a new cleavable linker-payload system enables stable drug delivery to cancer cells, causing selective destruction. HER2-targeting ADC trastuzumab deruxtecan demonstrated promising responsiveness in patients with HER2-positive CRC, in a phase 2 clinical trial (objective response rate = 45.3%). Furthermore, HER3-targeting patritumab deruxtecan, another ADC, exhibited impressive tumor shrinkage in pretreated patients with EGFR-mutated NSCLC, in a phase 1 clinical trial. This manuscript presents an overview of the accumulated evidence on HER2- and HER3-targeting therapy, especially ADCs, and discussion of remaining issues for further improving these treatments in cancers resistant to EGFR inhibitors.

## 1. Introduction

Aberrant overexpression of the epidermal growth factor receptor (EGFR) has been detected in various malignancies, including nonsmall cell lung cancer (NSCLC), colorectal cancer (CRC), and head and neck squamous cell carcinoma (HNSCC), and is associated with a poor prognosis [1,2,3]. The EGFR can be activated by its ligands, including EGF, transforming growth factor-α (TGF-α), amphiregulin, betacellulin, heparin-binding EGF, or epiregulin [4]. These ligands bind to the extracellular region of the EGFR, inducing a conformational change in EGFR, and leading to the dimerization and activation of EGFR signaling [4]. Some malignant cells aberrantly produce EGFR ligands, activating their own EGFRs in an autocrine fashion [5]. EGFR activation provokes the downstream activation of pathways, including ERK and AKT, leading to cancer cell survival and proliferation. EGFR inhibitors, including EGFR-tyrosine kinase inhibitors (TKIs) and anti-EGFR antibodies, can suppress cancer cell proliferation in NSCLC, CRC, and HNSCC, expressing aberrant EGFR ligands [5]. Furthermore, when treated with anti-EGFR antibodies, patients with high amphiregulin-expressing CRC have longer progression-free survival than those with low amphiregulin-expressing CRC [6]. Alternatively, EGFR is constitutively activated by its own mutations, including amino acid deletions and substitutions [7,8]. Somatic EGFR-activating mutations are observed in NSCLC, especially among women, nonsmokers, and Asian patients with adenocarcinoma [9].

EGFR-inhibitors include EGFR-TKIs, a standard therapy for EGFR-mutated NSCLC, and anti-EGFR antibodies, a standard therapy for CRC, HNSCC, and squamous cell lung cancer [10]. EGFR-TKIs compete with ATP to bind to the intracellular kinase domain of EGFR [7,8], whereas anti-EGFR antibodies bind to the extracellular region of EGFR preventing the binding of ligands to EGFR [5]. Both of these EGFR inhibitors restrain EGFR and its downstream activation in cancer cells, preventing cell proliferation and causing apoptosis. EGFR-inhibitors can shrink or stabilize tumors for a limited period because all tumors eventually acquire resistance to these inhibitors. Additionally, a subpopulation of cancer is primarily resistant to EGFR inhibitors. In EGFR-mutated NSCLC treated with EGFR-TKIs, the underlying mechanism of its adaptive resistance is largely a secondary EGFR mutation that causes amino acid substitutions, such as T790M and C797S, in an intrinsic kinase domain [11,12]. These mutations are referred to as gatekeeper mutations, which hinder the binding of TKIs to the kinase domain, rendering cancer cells insusceptible to TKIs. EGFR-secondary mutation, T790M, is observed in approximately 50% of patients with EGFR-mutated NSCLC who were treated with first or second generation EGFR-TKIs [13]. On the other hand, resistance to anti-EGFR antibodies is not associated with EGFR mutations in their kinase domains. Instead, resistance to these agents is frequently caused by genomic mutations of RAS or BRAF, located downstream of EGFR and that spontaneously activate themselves, even in the presence of anti-EGFR antibodies [14]. In CRC treated with anti-EGFR antibodies, RAS and BRAF genomic mutations were observed in approximately 40% and 10% cases, respectively [15].

## 2. HER Family-Mediated Resistance to EGFR-Inhibitor

### 2.1. HER2-Mediated Resistance to EGFR-Inhibitors in CRC and EGFR-Mutated NSCLC

HER2 is another proto-oncogene and its genomic amplification or mutation is a dominant mechanism for oncogenesis in certain malignancies. HER2 genomic amplification frequently appears in breast (BC) and gastric cancer (GC), with an incidence of approximately 25% and 15%, respectively; whereas it is relatively uncommon in NSCLC and CRC, with an incidence of approximately <5% [16]. However, HER2 genomic amplification was unexpectedly discovered in NSCLC and CRC, following their development of resistance to EGFR-inhibitors, such as anti-EGFR antibody cetuximab [17,18]. Furthermore, HER2-targeting agents or siRNA could restore the sensitivity to cetuximab in these resistant cancer cells, validating that HER2 contributed to cetuximab resistance (Figure 1). Strikingly, HER2 genomic amplification was detected in tumors obtained from patients with metastatic CRC, who developed an adaptive resistance to cetuximab-based therapy, whereas it was not detected in archival tumor samples [17,19]. Bertotti et al. also found HER2 genomic amplification in patient-derived xenografts (PDX) of CRCs resistant to anti-EGFR antibody treatment [18]. Notably, the incidence of HER2 genomic amplification was enriched in KRAS wild-type tumors, observed in approximately 13.6% of all KRAS wild-type tumors, whereas its frequency was 2.7% in the genetically unselected population, suggesting the mutual exclusivity between HER2 amplification and KRAS mutations for exerting resistance to anti-EGFR antibody [18]. HER2 genomic examination by Takezawa et al. revealed that it was amplified in around 12% of NSCLC with acquired resistance to EGFR-TKIs versus in only 1% of untreated lung adenocarcinomas, which is mutually exclusive for the EGFR secondary mutation T790M [20].

Our investigation of the underlying mechanism of this resistance suggested that an amplified HER2 causes aberrant activation, leading to bypassing EGFR-ERK to HER2-ERK signaling transduction [17]. Specifically, the anti-EGFR antibody cetuximab inhibits EGFR and its downstream ERK activation, which hinders cancer cell proliferation, whereas ERK activation is maintained in HER2-amplified cells, even under cetuximab treatment (Figure 1). HER2-specific inhibition with siRNA or HER2 inhibitors, such as lapatinib, results in the blockade of HER2 and ERK activation in these HER2-amplified cells, leading to the restoration of cetuximab susceptibility [17]. These results suggest that HER2-mediated bypassing signal causes resistance to EGFR inhibitors; therefore, HER2 blockade agents could potentially exert anticancer efficacy or recover the susceptibility to EGFR inhibitors in patients with HER2-amplified cancers. Indeed, Bertotti et al. demonstrated that dual targeting of HER2 and EGFR, such as a combination of lapatinib and pertuzumab or lapatinib and cetuximab, induced tumor regression in a cetuximab-resistant, HER2-amplified CRC PDX mouse model [18]. Finally, clinical trials for HER2-targeting agents in patients with HER2-amplified CRC revealed tumor shrinkage in anti-EGFR antibody-resistant tumors [21]. Specifically, the combination of trastuzumab and lapatinib (HERACLES trial) resulted in 8 of 27 (30%) patients achieving an objective response and 12 (44%) patients with stable disease [21].

The identification of HER2 genomic amplification in cancers as a cause of EGFR-inhibitor resistance has become a promising therapeutic target, especially in HER2-amplified CRC. However, the limited responsiveness to HER2-signaling blockade may require a distinct treatment strategy for conquering the resistance to EGFR inhibitors.

### 2.2. HER3-Mediated Resistance to EGFR-Inhibitors in Cancer

HER3 is another member of the HER family and is aberrantly expressed in a large number of malignancies, including BC, GC, CRC, and NSCLC [22]. HER3 has attracted less attention from researchers than EGFR or HER2, and there were no approvals for HER3 targeting agents. However, accumulated evidence indicates that HER3 plays a crucial role in cancer cell survival and drug resistance, especially in EGFR- or HER2-targeting therapy in certain malignancies [17,22,23,24,25,26]. HER3 binds to the neuregulin family, which comprises EGF-like family ligands, including heregulin, and alters the steric structure of its extracellular domain, causing HER3 activation by coupling with other receptors [22]. Unlike the other HER family members, HER3 has impaired kinase activity that precludes HER3 autophosphorylation; however, it could gain its transphosphorylation through heterogeneous coupling with various receptors, including EGFR, HER2, HER4, and MET [22]. The intracellular domain of HER3 has six docking sites for the p85 subunit of PI3K. The binding of HER3 with the p85 subunit of PI3K subsequently provokes PI3K/AKT downstream signaling, allowing cancer cells to be antiapoptotic [22].

In EGFR-mutated NSCLC, HER3 preferentially couples with EGFR and potently activates the PI3K/AKT signaling pathway to maintain antiapoptosis [27]. When EGFR-TKIs block EGFR activation in these cells, HER3 and its downstream pathway becomes inactive, inducing apoptosis. In addition to its involvement in EGFR-TKI susceptibility, HER3 also plays a key role in EGFR-TKI resistance in EGFR-mutated NSCLC. Specifically, when EGFR-mutated NSCLC cells have coexisting MET genomic amplification, these cells could maintain the antiapoptotic HER3/PI3K/AKT signaling, even under EGFR-TKI treatment [27] through HER3 coupling with MET instead of HER3/EGFR coupling to maintain HER3 activation. Indeed, HER3 repression by shRNA decreases the viable cell count in these MET-amplified cells, suggesting that these cells still depend on HER3 for survival [27].

HER3 also contributes to drug resistance via upregulation of HER3 in cancer cells resistant to EGFR-TKIs. Specifically, Sergina et al. observed that a cell membrane HER3 is upregulated in persistent BC cells after treatment with HER-family TKI, and that HER3-targeting siRNA could enhance susceptibility to EGFR-TKI in these cells [28]. They suggested that cell membrane HER3 upregulation by HER-family TKI causes a compensatory shift in HER3 phosphorylation–dephosphorylation equilibrium, by driving the phosphorylation reaction and reducing HER3 phosphatase activity. This HER3 alteration sustains antiapoptotic HER3/PI3K/AKT signaling. Haikala et al. reported that EGFR-mutated NSCLC PDX models also demonstrated an upregulated cell membrane HER3 expression and EGFR/HER3 dimerization after treatment with EGFR-TKI osimertinib [29]. Finally, we observed HER3 upregulation in EGFR-mutated NSCLC cell lines treated with EGFR-TKIs and in tumor samples obtained from patients with EGFR-mutated NSCLC treated with EGFR-TKIs [30,31] (Figure 1).

Collectively, HER3 expression could contribute to EGFR-TKI resistance in EGFR-mutated NSCLC by maintaining antiapoptotic HER3/PI3K/AKT signaling. Although the underlying mechanisms causing this aberrant expression of HER3 in cancer cells, especially following EGFR-targeting treatment, is not fully elucidated, HER3 may be a potential target for anticancer therapy in these resistant cells. Furthermore, given that HER3 is highly expressed in other malignancies, including CRC, it may become a treatment target across varied malignancies [24].

### 2.3. Heregulin Is Potentially Involved in HER3-Mediated Insensitivity to EGFR-Inhibitors in Cancer

Heregulin is a member of the neuregulin family that binds to HER3 and HER4 and is aberrantly expressed in certain cancer cells or stromal cells in tumors [22]. Heregulin is produced in the intercellular matrix and aberrantly activates HER3 in cancer cells in an autocrine or paracrine manner. Heregulin expression levels vary across multiple types of cancer, with HNSCC and squamous cell lung cancer expressing high levels [22]. Heregulin causes insensitivity to EGFR-tyrosine kinase inhibitors, which are anticancer agents [17,23,24,32] (Figure 1). We observed that the heregulin mRNA level is significantly greater in anti-EGFR antibody cetuximab-resistant HNSCC FaDuCR cells than in cetuximab-sensitive FaDu cells [32]. Furthermore, siRNA targeting heregulin in cetuximab-resistant FaDuCR cells reduced HER3 phosphorylation and enhanced susceptibility to cetuximab. Alternatively, heregulin genomic induction increases the phosphorylation level of HER3, providing resistance to EGFR-inhibitors in EGFR-mutated NSCLC or CRC cells [23,24]. Consistent with these preclinical examinations, clinical observations suggest a relationship between heregulin and insusceptibility to EGFR-inhibitors. Therefore, as observed in patients with CRC who were treated with cetuximab-based therapy, heregulin mRNA expression level was adversely related to progression-free survival (PFS) [17]. Alternatively, a soluble form of heregulin in plasma is associated with shorter PFS to EGFR-TKI treatment in patients with EGFR-mutated NSCLC [33]. However, given that this is a retrospective analysis, the relationship between heregulin and EGFR-TKI insensitivity remains clinically controversial and requires further validation.

## 3. HER3-Targeting Monoclonal Antibody Strategy: Its Potential and Limitations

Various monoclonal anti-HER3 antibodies have been investigated in NSCLC, CRC, HNSCC, and BC [22,23]. Preclinical studies have generally suggested that anti-HER3 monoclonal antibodies deliver anticancer efficacy, depending on heregulin expression in cancer cells. Several (but not all) of anti-HER3 antibodies bind with extracellular domain of HER3 that is competitive with heregulin; thereby, clinical trials of these antibodies focused on heregulin expressing cancers. Regarding anti-HER3 antibody patritumab, which is a fully human monoclonal immunoglobulin G1 (IgG1) antibody, our in vitro cell line-based screening using 48 NSCLC cell lines suggested that the anticancer efficacy of patritumab depends on the level of heregulin expression rather than that of HER3 expression [23]. Similarly, a xenograft mouse model study also observed the anticancer efficacy of patritumab in highly heregulin-expressing NSCLCs. Additionally, patritumab could restore susceptibility to EGFR-TKI in heregulin-expressing, EGFR-mutated NSCLC PC9HRG cells [23]. Based on these results, a phase II clinical trial for patritumab plus erlotinib, an EGFR-TKI, was conducted with a prespecified heregulin biomarker, in patients with locally advanced or metastatic NSCLCs. Patients with high heregulin mRNA expression in tumors had significantly better PFS with patritumab plus erlotinib than PFS with placebo plus erlotinib [34]. Similarly, those with a high level of soluble heregulin in plasma also had better PFS with patritumab plus erlotinib than the placebo arm [35]. However, this study could not prospectively determine the cutoff point for high heregulin. A subsequent randomized clinical trial for patritumab plus erlotinib was terminated at an interim analysis because the patritumab arm did not show any clinical benefit compared to the placebo arm, in patients with locally advanced or metastatic NSCLC with high heregulin expression [36]. Similar to patritumab, the efficacy of other anti-HER3 monoclonal antibodies was detected preclinically in some cancer cells across various cancers, especially in those with a high expression level of heregulin, but the clinical trials conducted for these antibodies could not find any clinical significance in NSCLC, BC, colon, and even HNSCC, which has the highest expression level of heregulin among malignancies [24]. This discrepancy between the preclinical and clinical anticancer efficacy of anti-HER3 antibody has not been elucidated. Given that patritumab could delay tumor growth but not shrink those tumors in a xenografted mouse model study, anti-HER3 antibodies may require drug conjugation or carbohydrate chain modification to enhance their anticancer efficacy.

Another limitation of anti-HER3 monoclonal antibody is that it could not necessarily block HER3 activation, depending on the mutated-EGFR. Specifically, patritumab did not alter HER3 phosphorylation levels in EGFR-mutated NSCLC cells, such as HCC827 cells, and had no anticancer effect on these cells [23]. Consistently, a clinical trial for patritumab combined with erlotinib did not observe any response in patients with EGFR-mutated NSCLC, previously treated with EGFR-TKIs [37]. Patritumab binds to the extracellular domain of HER3, and presumably, prevents its binding with ligands, such as heregulin, whereas it probably does not block HER3/EGFR coupling.

Overall, the strategy of HER3-targeting using monoclonal antibodies may be insufficient for treating EGFR-mutated NSCLC. The conceivable resolution for this limitation may be antibody modulation, such as drug conjugation. 

## 4. Pan-HER Family Tyrosine Kinase Inhibitors Have Shown Superior Efficacy in Cancers Resistant to EGFR-Inhibitors

Meanwhile, pan-HER kinase inhibitors, such as afatinib, possibly have superior efficacy compared to EGFR-inhibitors. They could block EGFR, HER2, and HER4 kinase, along with preventing HER3 transphosphorylation, mediated by other HER family members [38,39]. Such a unique characteristic may be advantageous in treating EGFR-mutated NSCLCs, especially those expressing heregulin. Furthermore, we observed that heregulin genomic induction potently phosphorylates HER2, HER3, and HER4, in addition to EGFR, in EGFR-mutated NSCLC PC9HRG cells, as heregulin leads to the preferential coupling of HER3 with HER2 [38]. EGFR-TKI erlotinib definitely decreases the phosphorylation level of EGFR; however, it cannot adequately decrease the phosphorylation of other HER family members in these cells, thereby maintaining downstream activation, such as ERK and AKT. Meanwhile, the pan-HER kinase inhibitor, afatinib thoroughly decreases the phosphorylation levels of pan-HER members, including EGFR, HER2, HER3, and HER4, and decreases the phosphorylation of their downstream pathways, ERK and AKT [38]. Consistently, heregulin-expressing PC9HRG xenografted mouse study observed shrinkage in tumors treated with afatinib, although no such changes were observed in tumors treated with erlotinib [38]. Furthermore, other investigators reported a combination of three anti-HER family drugs, cetuximab (an anti-EGFR mAb), trastuzumab (an anti-HER2 mAb), and osimertinib (EGFR-TKI), as an effective and long-lasting treatment that is able to prevent onset of resistance to osimertinib in their mouse model study [40]. This result suggested that multiple HER family blockade had superior efficacy compared to the EGFR-inhibitors. Finally, a clinical trial for afatinib in locally advanced or metastatic EGFR-mutated NSCLC revealed a significantly longer PFS with afatinib treatment than with gefitinib (mPFS, 11.0 months vs. 10.9 months; hazard ratio (HR), 0.73; 95% confidence interval (CI), 0.57–0.95; *p* = 0.017) [41]. Dacomitinib, another pan-HER kinase inhibitor, significantly prolonged PFS for patients with EGFR-mutated NSCLC, compared to gefitinib (mPFS, 14.7 months vs. 9.2 months; hazard ratio (HR), 0.59; 95% confidence interval (CI), 0.47–0.74; *p* < 0.0001) [42].

Pan-HER kinase inhibitors, such as afatinib and dacomitinib, are more effective than EGFR-TKIs, such as gefitinib and erlotinib, especially in EGFR-mutated NSCLCs. However, pan-HER kinase inhibitors are more toxic, with side effects, such as diarrhea and rash, than EGFR-inhibitors. Therefore, it is desirable to distinguish the subpopulation that has an advantage with pan-HER kinase inhibitors, as compared to EGFR-inhibitors, by using a predictive biomarker. 

## 5. Novel Antibody Drug Conjugates (ADCs) for Anticancer Treatment

### 5.1. HER2-Targeting ADC Trastuzumab Deruxtecan and Its Application for Conquering Resistance to EGFR-Inhibitors

ADC development is based on the concept of “a magic bullet,” established a century ago, implying the selective delivery of toxic agents to the crux of diseases [43]. ADCs consist of three components: an antibody, a payload with cytotoxic potency, and a linker for connecting them (Figure 2). ADCs form complexes with antigens on the cell membrane, following which are then internalized and transferred to lysosomes, wherein ADCs are digested, thereby releasing the payloads [43]. A payload is usually a cytotoxic chemical compound, such as emtansine, which inhibits tubulin polymerization. Since the late 1980s, clinical trials for early generation ADCs have been conducted, and their toxicity and poor efficacy has been frequently observed. A possible reason for its poor efficacy was the fact that chimeric antibody from humans and mice could not stabilize circulating ADCs, which easily released unbound payloads into the system [43]. However, the use of humanized monoclonal antibodies, such as trastuzumab or other fully human ones, remarkably improved the half-life of ADCs in the blood, which enabled safe and efficient delivery of ADCs to cancer cells because of their excellent stability. For example, anti-HER2 ADC trastuzumab emtansine (T-DM1) demonstrated responsiveness and well-manageable toxicity in patients with BC, who were resistant to other HER2-targeting therapies, including trastuzumab and pertuzumab; it finally demonstrated significantly prolonged survival as compared to standard therapies, in patients with HER2-positive BC [44]. Additionally, several ADCs targeting other antigens, including CD30 and CD22, have been approved for use in multiple malignancies.

However, drug resistance is a critical issue concerning ADC treatment. Patients with HER2-positive BC eventually acquire resistance to T-DM1. Our preclinical studies indicated that high expression of ATP-binding cassette (ABC) transporters, such as ABCC2 and ABCG2, causes resistance by efflux of its payload, emtansine, from cells [45]. Alternatively, antigen loss, including HER2, can cause resistance to ADCs. Loganzo et al. subjected cells of the HER2-positive BC line JIMT1 to T-DM1 to induce resistance to this ADC in vitro. These resistant cells exhibited a marked decrease in HER2 protein expression after several months of T-DM1 exposure [46].

Finally, the next generation of anti-HER2 ADC trastuzumab deruxtecan solved this drug resistance. According to Ogitani et al., trastuzumab deruxtecan loads a newly developed cleavable linker that connects trastuzumab and the payload [47]. A highlight of the new technology utilized in trastuzumab deruxtecan is that this linker is remarkably stable in systemic circulation owing to its excellent hydrophilicity, and is selectively digested by cathepsin B in the lysosome of cancer cells [43,47]. This stable linker system reduces an unbound payload in systemic circulation and allows the intensive drug delivery to cancer cells [48]. In general, the drug-to-antibody ratio (DAR) of ADCs is limited (that of T-DM1 is 3.5) because greater DARs tend to result in unstable ADCs with higher clearance rates, which can limit anticancer efficacy and increase systemic toxicity [43,47]. However, the newly developed linker system employed in trastuzumab deruxtecan can increase its DAR to approximately eight, owing to its excellent stability [47]. Collectively, the stability of trastuzumab deruxtecan in plasma demonstrates slower clearance and achieves greater intracellular delivery of its payload to cancer cells, when compared to that of T-DM1 [47,48]. Another unique structural characteristic of trastuzumab deruxtecan is that its payload is a novel exatecan-derivative topoisomerase I inhibitor (DXd), which could potently damage DNA, leading to apoptosis [48]. Our preclinical study using T-DM1-resistant HER2-positive cell line observed noncross resistance between T-DM1 and trastuzumab deruxtecan because DXd may be a poor substrate of ABC transporters, such as ABCC2 and ABCG2, in comparison to emtansine [45].

In addition to its highly selective and potent efficacy, trastuzumab deruxtecan uniquely has a “bystander antitumor effect,” which includes anticancer effects for not only antigen-expressing tumor cells, but also adjacent antigen-negative or less-expressing cells, through the transfer of released payload from the antigen-expressing cells to the neighboring antigen-negative or less-expressing cells [49,50]. This is an advantage for treating tumor-expressing antigens heterogeneously, such as HER2-positive GC. In a phase 2 clinical study (DESTINY Gastric-01, NCT03329690), trastuzumab deruxtecan significantly improved overall survival (OS), when compared to physician’s choice of irinotecan or paclitaxel chemotherapy, in patients with previously treated HER2-positive GC (median, 12.5 vs. 8.4 months; HR for death, 0.59; 95% CI, 0.39–0.88; *p* = 0.01); however, T-DM1 did not demonstrate improved OS in this target [51,52]. Trastuzumab deruxtecan is currently approved for treating patients with HER2-positive BC who are treated with other HER2-targeting agents or patients with HER2-positive GC, in Japan [51,53].

Strikingly, trastuzumab deruxtecan demonstrated responsiveness in multiple HER2-positive cancers, including CRC and NSCLC, which are assumed to be resistant to EGFR-inhibitors [54] (Figure 2). According to the results of the phase II DESTINY-CRC01 study conducted among patients with advanced HER2-positive CRC (3+ IHC or IHC 2+/in situ hybridization +), 45% of heavily pretreated patients responded to this single agent, whereas 83% achieved disease control [55]. Similar responses were seen in patients with other prior anti-HER2 treatment (43.8%) and in those with no prior anti-HER 2 treatment (45.9%). Therefore, patients with HER2-positive CRCs may gain limited benefits from EGFR inhibitors, although this could be achieved by treatment with trastuzumab deruxtecan.

Additionally, according to interim results in a clinical trial of HER2-overexpressing or mutated NSCLC (the phase II DESTINY-Lung01 trial), trastuzumab deruxtecan demonstrated favorable clinical activity, with a high objective response rate (ORR) of 61.9% and favorable PFS of 14 months, in cohort 2, in the HER2-mutated population [56]. Cohort 1, which included HER2-overexpression [IHC 3+ or IHC 2+], in the same trial is ongoing, and the efficacy of trastuzumab deruxtecan for patients with HER2-overexpressing NSCLC has yet to be determined.

### 5.2. HER3-Targeting ADC Patritumab Deruxtecan for EGFR-Mutated NSCLC Resistant to EGFR-TKIs

HER3-targeting patritumab deruxtecan is another ADC using the same linker-payload system as trastuzumab deruxtecan, which is conjugated with anti-HER3 monoclonal antibody patritumab [30,57]. According to a preclinical study using MDA-MB-453 cells, expressing both HER2 and HER3, conducted by Hashimoto et al., patritumab deruxtecan had a lower binding level than anti-HER2 monoclonal antibody (mAb) [57]. However, patritumab deruxtecan was efficiently internalized into cells with a higher internalization rate than anti-HER2 mAb, reaching 64.1% and 1.7% at 1 h for patritumab deruxtecan and anti-HER2 mAb, respectively [57]. After the internalization of patritumab deruxtecan, a linker cleavage releases payloads, causing cancer cells to undergo apoptosis initiated through DNA damage. Hashimoto et al. showed that patritumab deruxtecan could block HER3/PI3K/AKT signaling in heregulin-expressing cancer cells, whereas it caused apoptosis in HER3-expressing cancer cells, regardless of heregulin expression, suggesting that the apoptotic effect is mainly attributed to DNA damage by payload [57]. Furthermore, through the study of various xenograft models conducted by Hashimoto et al., HER3 expression is an important determinant of tumor regression caused by patritumab deruxtecan treatment [57].

In EGFR-mutated NSCLC, we observed that patritumab deruxtecan exerts its anticancer efficacy depending on the level of HER3 expression, which is maintained or even increased after acquiring resistance to EGFR-TKIs [30] (Figure 2). Specifically, EGFR-mutated and MET-amplified NSCLC HCC827GR5 cells are resistant to EGFR-TKIs; however, they are still sensitive to patritumab deruxtecan, similar to the parental HCC827 cells, because HCC827GR5 cells maintain a high level of HER3 expression on the cell surface, similar to HCC827 cells [30]. Osimertinib is third generation EGFR-TKI and exerts anticancer activity in NSCLC harboring EGFR-secondary mutation T790M [58]. Osimertinib-resistant PC9AZDR7 cells express three times greater HER3 expression on the cell surface than parental EGFR-mutated PC9 cells, and are more susceptible to patritumab deruxtecan than the parental PC9 cells [30]. Interestingly, tissue samples obtained from patients with EGFR-mutated NSCLC showed greater HER3 expression after becoming EGFR-TKI resistant than that before treatment [31].

Consistent with our preclinical study, the phase I clinical trial for patritumab deruxtecan in patients with EGFR-mutated NSCLC demonstrated an objective response rate of 25% and a disease control rate of 70% [59]. Although the median follow-up time was only 5 months (range, 0–15 months) and 28 of 57 patients (49%) are still undergoing treatment, its responsiveness is impressive in patients who were heavily pretreated with a median number of four regimens, including EGFR-TKIs, and its toxicity is well-tolerable [59]. Notably, its efficacy is observed regardless of the resistance mechanisms for EGFR-TKIs, including C797S secondary EGFR mutation, MET amplification, HER2 mutation, BRAF fusion, and PIK3CA mutation [59]. Generally, the EGFR-TKI resistance mechanism is quite heterogeneous among EGFR-mutated cancers; therefore, patritumab deruxtecan may have an advantage, for treating such a heterogeneous population, over other agents, such as MET-inhibitors. Concerning the toxicity of patritumab deruxtecan, the most common ≥ grade 3 treatment emergent adverse events (TEAEs) were thrombocytopenia (25%) and neutropenia (16%) [59]. There were no treatment-related TEAEs associated with fatality. Collectively, patritumab deruxtecan demonstrated a manageable safety profile and clinically meaningful antitumor activity at 5.6 mg/kg (the recommended dose for expansion). The remaining unresolved question is whether HER3 expression level correlates with the efficacy of patritumab deruxtecan. Alternatively, researchers may consider other clinical or molecular characteristics to predict its efficacy. Currently, a phase 2 study of single-agent patritumab deruxtecan is being planned for patients, following failure of EGFR TKIs and platinum-based chemotherapy.

### 5.3. Combination Strategy for HER3-Targeting ADC Patritumab Deruxtecan with EGFR-TKI or Anti-PD1/PD-L1 Antibody

To enhance the efficacy of patritumab deruxtecan, a combination strategy should be considered. For instance, our previous study observed that the anticancer efficacy of patritumab deruxtecan was enhanced after combining it with EGFR-TKI in EGFR-mutated NSCLC because EGFR-TKIs are known to upregulate HER3 expression on the surface of these cells [30]. Furthermore, according to Haikala et al., the upregulation of HER3 by EGFR-TKIs accelerates the internalization of patritumab deruxtecan in cancer cells, causing potent apoptosis by DNA damage [29]. Consistently, our clinical observation indicated that cell surface HER3 expression is upregulated in tumors with EGFR-mutation being treated with EGFR-TKI therapy [31]. Collectively, these results suggest that combining EGFR-TKIs with patritumab deruxtecan may strengthen the anticancer effects of patritumab deruxtecan in patients with EGFR-mutated NSCLC. Currently, a phase 1 clinical study is evaluating patritumab deruxtecan combination with osimertinib, as both first- and second-line treatment, in patients with advanced or metastatic EGFR-mutated NSCLC (NCT04676477).

Immune-oncology (IO) therapy, especially anti-PD1/PD-L1 antibody, has become a standard therapy for NSCLC; however, EGFR-mutated NSCLC is less susceptible to IO than other NSCLCs, and their tumor microenvironment (TME) has inactive tumor-infiltrating lymphocytes [60]. Haratani et al. showed that patritumab deruxtecan treatment improved the TME and caused immunogenic cell death in tumors, in a syngeneic mouse model study [61]. Specifically, selective and abundant supply of payload DXd, a topoisomerase I inhibitor, to tumors induced greater lymphocyte infiltration and repressed M1 macrophages in tumors [61]. This is caused by patritumab deruxtecan, which enhances the initiation of immunological reactions and antigen recognition. Furthermore, patritumab deruxtecan combination therapy with anti-PD1 antibody demonstrated a more potent immunologic reaction along with antitumor efficacy [61]. Although EGFR-mutated NSCLC has a less immunogenic microenvironment, patritumab deruxtecan may possibly improve it, enhancing the efficacy of anti-PD-1/PD-L1 therapy.

## 6. Conclusions

In the last two decades, EGFR-inhibitors have delivered tremendous benefits to patients with EGFR-mutated NSCLC, CRC, and HNSCC. Currently, the challenge of overcoming resistance to EGFR-inhibitors remains unresolved. Several resistance mechanisms for EGFR-inhibitors have been elucidated; HER2 and HER3 contribute to the induction of resistance to EGFR inhibitors in certain cancers. Targeting HER2 with trastuzumab deruxtecan and targeting HER3 with patritumab deruxtecan, using the newly evolved ADC technology, is now breaking this barrier. Monotherapy with these ADCs have presented impressive clinical outcomes, and preclinical studies are expecting an enhanced efficacy of patritumab deruxtecan, in combination with EGFR-TKI or anti-PD-1/PD-L1. An issue that remains concerns the identification of biomarkers for predicting the anticancer efficacy of ADCs, especially patritumab deruxtecan, whereas ongoing clinical trials will answer this question and clarify their advantages in patients with cancers resistant to EGFR-inhibitors.

## Figures and Tables

**Figure 1 cancers-13-01047-f001:**
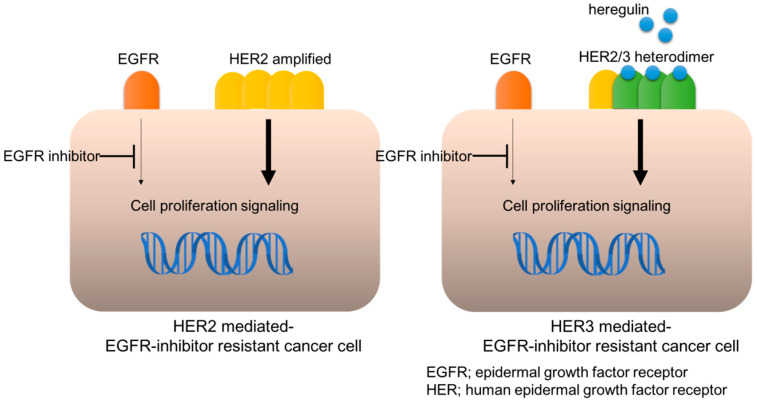
HER2- or HER3-mediated EGFR-inhibitor resistance.

**Figure 2 cancers-13-01047-f002:**
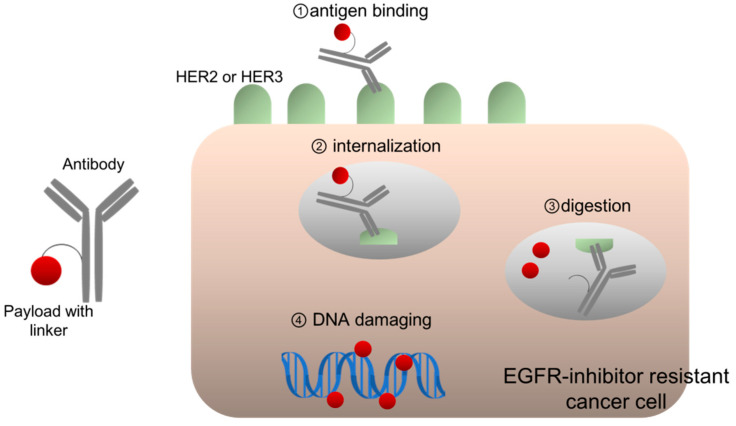
HER2- or HER3-targeting antibody-drug conjugates.

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
