# Peer review of "HER2-/HER3-Targeting Antibody—Drug Conjugates for Treating Lung and Colorectal Cancers Resistant to EGFR Inhibitors"

_cancers, 2021, doi:10.3390/cancers13051047_

Round 1

Reviewer 1 Report

This manuscript is very well written and the contents give a very clear picture of the involvement of the EGFR receptor family members in tumors and their response/resistance to target therapypies

Author Response

Dear reviewer 1

Thank you for your kind reviewing and comments.

Sincerely yours,

Kimio Yonesaka

Reviewer 2 Report

This is very good review of the literature with some insight into future directions.  Only a few minor issues should be addressed.  

I found no mention of glioblastoma and the role of EGFR-family members for this cancer.

The target of osimertinib, the T790M, needs to be highlighted.  This drug reacts preferentially with this mutant form of EGFR.

Finally, several (but not all) of the antibodies mentioned compete for ligand binding.  This information should be provided to the reader with details about potential rationale for the original development.

Author Response

Dear reviewer 2

Thank you for your kind comments.

#1 "I found no mention of glioblastoma and the role of EGFR-family members for this cancer."

Thank you for your comment. Previously I contacted with editorial office and we confirmed that this review paper focused on NSCLC and CRC.

#2 "The target of osimertinib, the T790M, needs to be highlighted.  This drug reacts preferentially with this mutant form of EGFR."

Thank you for your suggestion. I described this point at Page 8, line 35 as below.

Osimertinib is third generation EGFR-TKI and exerts anti-cancer activity in NSCLC harboring EGFR-secondary mutation T790M [58].

And I added a reference 

58. Mok, T. S.; Wu, Y. L.; Ahn, M. J.; Garassino, M. C.; Kim, H. R.; Ramalingam, S. S.; Shepherd, F. A.; He, Y.; Akamatsu, H.; Theelen, W.S.; et al. Mok, Tony S., et al. Osimertinib or platinum–pemetrexed in EGFR T790M–positive lung cancer. New England Journal of Medicine 2017, 376, 629-640.

#3. "Finally, several (but not all) of the antibodies mentioned compete for ligand binding.  This information should be provided to the reader with details about potential rationale for the original development."

Thank you for your comments. I described this point at Page 5, line 14 as below.

Several (but not all) of Anti-HER3 antibodies bind with extracellular domain of HER3 that is competitive with heregulin; thereby clinical trials of these antibodies focused on heregulin expressing cancers. 

Sincerely yours,

Kimio Yonesaka

Reviewer 3 Report

The review titled “ HER2-/HER3- targeting antibody-drug conjugates for treating lung and colorectal cancers resistant to EGFR inhibitors” provided very useful information about enhancing the efficacy of EGFR inhibitor through HER2-/HER3- targeting antibody in lung and colorectal cancers which are resistant to EGFR inhibitors. Firstly, it analyzed the role of HER2/HER3 in resistance to EGFR inhibitor, especially high lighting the potent influence of HER3 and its ligand , Heregulin, in EGFR inhibitor resistance. Secondly, authors described the potential of HER3-targeting monoclonal antibody strategy and its limitation. At last, they discussed the novel antibody drug conjugates(ADCs) for anti-cancer treatment, including the combination strategy for ADC with EGFR TKI or anti-PD1/PD-L1 antibody. This review was well organized and written. It’s easy to read and understood. It involved a relatively comprehensive and up-to-date analysis of HER family associated strategy in overcoming EGFR inhibitor resistance in lung and colorectal cancers. All these information are useful and helpful to the researcher in this field. However, there are some minor questions need to be considered.

1: the ref.22 used in discussion HER2-mediated resistance to EGFR inhibitors in CRC and EGFR-mutated NSCLS is inappropriate. This ref only discussed the combination of two HER2 targeting antibody. It was not associated with the topic , EGFR inhibitor resistance, here.

2: "the part 4: Pan-HER family tyrosine kinase inhibitors have shown....." , in this part the total ref is 2, please provide much and new studies to refer this topic.

3: in some units, the authors discussed studies from their own lab. It would be much better to involve others studies to support it.

Author Response

Dear reviewer 3

Thank you for your kind comments.

#1 " the ref.22 used in discussion HER2-mediated resistance to EGFR inhibitors in CRC and EGFR-mutated NSCLS is inappropriate. This ref only discussed the combination of two HER2 targeting antibody. It was not associated with the topic , EGFR inhibitor resistance, here."

Thank you for your suggestion. I deleted ref.22 and its related sentences in page 3. 

#2 "the part 4: Pan-HER family tyrosine kinase inhibitors have shown....." , in this part the total ref is 2, please provide much and new studies to refer this topic."

Thank you for your suggestion. I added three related ref for the part 4 as below.

39. Wang, X.; Batty, K. M.; Crowe, P. J.; Goldstein, D.; Yang, J. L. (2015). The potential of panHER inhibition in cancer. Frontiers in oncology 2015, 5, 2.

40. Romaniello, D.; Mazzeo, L.; Mancini, M.; Marrocco, I.; Noronha, A.; Kreitman, M.; Srivastava, S.; Ghosh, S.; Lindzen, M.; Salame, T.M.;et al. A combination of approved antibodies overcomes resistance of lung cancer to osimertinib by blocking bypass pathways. Clinical Cancer Research2018, 24, 5610-5621.

42. Wu, Y.L.; Cheng, Y.; Zhou, X.; Lee, K.H.; Nakagawa, K.; Niho, S.; Tsuji, F.; Linke, R.; Rosell, R.; Corral, J.; et al. Dacomitinib versus gefitinib as first-line treatment for patients with EGFR-mutation-positive non-small-cell lung cancer (ARCHER 1050): a randomised, open-label, phase 3 trial. The Lancet Oncology2017, 18, 1454-1466.

#3. "in some units, the authors discussed studies from their own lab. It would be much better to involve others studies to support it."

Thank you for your comments. I discussed studies from other group as described at Page 6, line 18 and Line 26 as below.

Furthermore, other investigators reported a combination of three anti-HER family drugs, cetuximab (an anti-EGFR mAb), trastuzumab (an anti-HER2 mAb), and osimertinib (EGFR-TKI), as an effective and long-lasting treatment that is able to prevent onset of resistance to osimertinib in their mouse model study [40]. 

And

Dacomitinib, another pan-HER kinase inhibitor, significantly prolonged PFS for patients with EGFR-mutated NSCLC, compared to gefitinib [mPFS, 14.7 months vs. 9.2 months; hazard ratio (HR), 0.59; 95% confidence interval (CI), 0.47–0.74; p < 0.0001] [42].

Sincerely yours,

Kimio Yonesaka
